# Universal Adversarial Attack using Very Few Test Examples

## Abstract

Adversarial attacks such as Gradient-based attacks, Fast Gradient Sign Method (*FGSM*) by Goodfellow et al. (2015) and *DeepFool* by Moosavi-Dezfooli et al. (2016) are input-dependent, small pixel-wise perturbations of images which fool state of the art neural networks into misclassifying images but are unlikely to fool any human. On the other hand a universal adversarial attack is an input-agnostic perturbation. The same perturbation is applied to all inputs and yet the neural network is fooled on a large fraction of the inputs. In this paper, we show that multiple known input-dependent pixel-wise perturbations share a common spectral property. Using this spectral property, we show that the top singular vector of input-dependent adversarial attack directions can be used as a very simple universal adversarial attack on neural networks. We evaluate the error rates and fooling rates of three universal attacks, SVD-Gradient, SVD-DeepFool and SVD-FGSM, on state of the art neural networks. We show that these universal attack vectors can be computed using a small sample of test inputs. We establish our results both theoretically and empirically. On VGG19 and VGG16, the fooling rate of SVD-DeepFool and SVD-Gradient perturbations constructed from observing less than 0.2% of the validation set of ImageNet is as good as the universal attack of Moosavi-Dezfooli et al. (2017a). To prove our theoretical results, we use matrix concentration inequalities and spectral perturbation bounds. For completeness, we also discuss another recent approach to universal adversarial perturbations based on $(p, q)$-singular vectors, proposed independently by Khrulkov & Oseledets (2018), and point out the simplicity and efficiency of our universal attack as the key difference.

## 1 Introduction

Neural network models achieve state of the art results on several image classification tasks. However, these models are known to be vulnerable to adversarial attacks. Szegedy et al. (2013) show that small, pixel-wise changes that are almost imperceptible to the human eye can make neural network models grossly misclassify images. Regarding an image as a vector $\mathbf{x}$ in $\mathbb{R}^d$, an adversarial, pixel-wise perturbation gives a small perturbation vector $\mathbf{v} \in \mathbb{R}^d$ to perturb $\mathbf{x}$ so that the network classifies $\mathbf{x} + \mathbf{v}$ differently from $\mathbf{x}$. Szegedy et al. (2013) find such a perturbation vector using box-constrained L-BFGS. Goodfellow et al. (2015) propose the Fast Gradient Sign Method (*FGSM*) as a faster approach to find such an adversarial perturbation. In subsequent work on *FGSM*, its iterative variant was introduced by Kurakin et al. (2017) and another version called Projected Gradient Descent (*PGD*) by Madry et al. (2018). In the works of Goodfellow et al. (2015) and Madry et al. (2018), the adversarial perturbations considered are from an $\ell_\infty$-ball around the input $\mathbf{x}$ - namely, each pixel value is perturbed by a quantity within $[-\epsilon, +\epsilon]$. DeepFool by Moosavi-Dezfooli et al. (2016) gives a method to compute a minimal $\ell_2$-norm perturbation to fool the input. To understand how adversarial attacks transfer across different models Tramer et al. (2017) also study model-agnostic perturbations using the direction of the difference between the intra-class means.

Universal adversarial perturbations are input-agnostic so that the same perturbation fools the trained model on a large fraction of test inputs. Recent work by Moosavi-Dezfooli et al. (2017a;b) on universal adversarial perturbations looks at the curvature of the decision boundary. Their universal attacks are more sophisticated than the simple and fast, gradient-based adversarial perturbations, and require significantly more computation. In Moosavi-Dezfooli et al. (2017b) the authors give

a theoretical analysis of existence of universal adversarial perturbations using directions in which the decision boundary has positive curvature. Khrulkov & Oseledets (2018) propose a different method to construct a universal adversarial perturbation. Their perturbation is based on computing the $(p, q)$-singular vectors of the Jacobian matrices of hidden layers of the network.

The above discussion raises an important question: are there any interesting properties shared by the input dependent adversarial attacks which can be exploited to come up with an easy to compute universal perturbation? Can one find such a universal perturbation using only a small fraction of test inputs and also ensure that a perturbation along this direction fools most test inputs? We answer both of these questions affirmatively in this paper. Furthermore, we show that our universal attack compares favourably with the universal attack in Moosavi-Dezfooli et al. (2017a).

## 2 Summary of our main results

- Our first observation is that even though the adversarial attack directions are input-dependent, overall they have only a small number of dominant principal components. We show this for attacks based on the *Gradient* of the loss function, the *FGSM* attack and DeepFool, on a variety of architectures and datasets.

- Construct a matrix whose $i$-th row is an input dependent attack vector on the $i$-th test point. Our second observation is that a small pixel-wise perturbation in the direction of the top principal component of this matrix alone can be used as a universal attack. This simple algorithm gives us SVD-Gradient, SVD-FGSM and SVD-DeepFool perturbations.

- Our third observation is that the top principal component can be well-approximated from a very small sample of the test data, and small samples are good enough to obtain SVD-Universal perturbations comparable to that in Moosavi-Dezfooli et al. (2017a).

- We give a theoretical justification of this phenomenon using matrix concentration inequalities and spectral perturbation bounds. This observation holds across multiple input-dependent adversarial attack directions given by *Gradient*, *FGSM* and *DeepFool*.

## 3 Definitions and the problem formulation

Assume that we have a distribution $\mathcal{D}$ on pairs of images and labels, with images coming from a set $\mathcal{X}$. In the case of CIFAR-10 for example we can think of $\mathcal{X}$ to be a set of all $32 \times 32$ images with pixel values from $[0, 1]$. Let $(X, Y)$ be a sample from $\mathcal{D}$ and let $f : \mathcal{X} \to [k]$, be a $k$-class classifier. The natural accuracy $\delta$ of the classifier $f$ is defined to be

$$\Pr_{(X,Y) \in \mathcal{D}}[f(X) = Y].$$

We regard a deterministic adversary to be a function $\mathcal{A} : \mathcal{X} \to \mathcal{X}$. The error rate of $\mathcal{A}$ on the classifier $f$ is defined to be

$$\Pr_{(X,Y) \in \mathcal{D}}[f(X + \mathcal{A}(X)) \neq Y].$$

When $\mathcal{A}$ is a distribution over functions, we get a randomized adversary. The norm of the perturbation applied to $X$ is the norm of $\mathcal{A}(X)$ (we only consider $\ell_2$ norm in this paper).

In Moosavi-Dezfooli et al. (2017a;b) and in Khrulkov & Oseledets (2018) the authors consider the fooling rate of an adversary and not the error rate. Recall that $f$ is said to be fooled on input $x$ by the perturbation $\mathcal{A}(x)$ if $f(x) \neq f(x + \mathcal{A}(x))$. And the fooling rate is defined to be the probability that $f$ is fooled by choosing $x$ at random. Note that

$$\begin{aligned}
&\Pr_{(X,Y) \in \mathcal{D}}[f(X + \mathcal{A}(X)) \neq Y] \\
&= \Pr_{(X,Y)}[f(X + \mathcal{A}(X)) \neq Y | f(X) = Y] \Pr_{(X,Y)}[f(X) = Y] \\
&+ \Pr_{(X,Y)}[f(X + \mathcal{A}(X)) \neq Y | f(X) \neq Y] \Pr_{(X,Y)}[f(X) \neq Y] \\
&\leq \Pr_{(X,Y)}[f(X + \mathcal{A}(X)) \neq Y | f(X) = Y] + (1 - \delta).
\end{aligned}$$

So, if the natural accuracy of the classifier is high, the fooling rate is close to the error rate.

$$\begin{aligned}
&\Pr_{(X,Y)}[f(X + \mathcal{A}(X)) \neq f(X)] \\
&\geq \Pr_{(X,Y)}[f(X + \mathcal{A}(X)) \neq Y] - (1 - \delta).
\end{aligned}$$

The error rate of the adversary with zero perturbation is the error rate of the trained network, whereas the fooling rate of the adversary with zero perturbation is necessarily zero. However, small fooling rate does not necessarily imply small error rate, especially when the natural accuracy is not close to 100%. Note that existing models such as VGG16, VGG19, ResNet50 do not achieve natural accuracy greater than 0.8.

## 3.1 INPUT DEPENDENT ATTACKS

There are a number of input dependent attacks known and our generic algorithm can be used to produce a universal adversary from each of them. We demonstrate the effectiveness of our method by applying it to a few popular input dependent attacks, which we discuss below. Let $L(\theta, x, y)$ denote the loss function used to train the model. Here $\theta$ are the model parameters, $x$ is the input and $y$ is the class label.

In the *Gradient* attack the adversarial example produced on input $x$ is

$$x' = x + \epsilon \nabla_x L(\theta, x, y).$$

In the *FGSM* attack proposed by Goodfellow et al. (2015) the adversarial example produced on input $x$ is

$$x' = x + \epsilon \operatorname{sign}\left(\nabla_x L(\theta, x, y)\right).$$

So the perturbed input is within an $\ell_\infty$ ball of radius $\epsilon$ about $x$.

*DeepFool* proposed by Moosavi-Dezfooli et al. (2016) is an iterative algorithm. At each stage of the iteration the region of space around the current perturbed input $x^t$ where the classifier outputs label $f(x^t)$ is approximated by a polyhedron $\mathcal{P}_t$. $x^t$ is projected to the closest face of $\mathcal{P}_t$. We stop at the smallest $t$ for which $f(x) \neq f(x^t)$.

## 3.2 EXISTING UNIVERSAL ATTACKS

In Moosavi-Dezfooli et al. (2017a), the authors fix a budget $\epsilon$, a bound on the allowed $\ell_2$ norm of the attack[1] and also the desired fooling rate. They pick a sample $S$ of the training data. Starting with the zero perturbation $v$, the algorithm proceeds in stages as long as the fooling rate of the current perturbation vector $v$ is less than the desired fooling rate on $S$. They run over $x \in S$, for which $f(x) = f(x + v)$. For each such $x$ they find the smallest perturbation $r_x$ sending $x$ to the boundary (using DeepFool) and $v$ is updated to $\Pi(v + r_x)$ where $\Pi$ is the projection onto the ball of radius $\epsilon$.

In Khrulkov & Oseledets (2018) the authors consider the outputs $f_i(x)$ of the $i$-th hidden layer of a network on an input $x$. Denote by $J_i(x)$ the matrix of partial derivatives $\frac{\partial f_i(x)}{\partial x}$ evaluated at $x$. They pick a sample $S$ of $\mathcal{X}$ of size $m$ and compute for each $x \in S$, $J_i(x)$. Stacking these in different rows of a matrix $J$, they use as their universal attack the vector $\operatorname{argmax}||Jv||_q$ subject to $||v||_p = 1$.

In Mopuri et al. (2017) the authors propose a universal attack which is completely data independent. It only depends upon the model architecture. Given a trained neural network with $K$ hidden (convolution) layers, the authors start with a random image $v$ and minimize $\Pi_{i=1}^{i=K} \bar{\ell}_i(v)$, $||v||_\infty \leq \epsilon$. Here $\bar{\ell}_i(v)$ is the mean activation of layer $i$ when the input is $v$. The authors show that the perturbation $v$ achieving the minimum is a data agnostic attack and achieves a reasonable fooling rate.

# 4 UNIVERSAL ADVERSARIAL ATTACK USING A SMALL SAMPLE

## 4.1 SVD-UNIVERSAL ALGORITHM

Algorithm 1 gives the pseudocode to construct a universal attack for a given model. The output of the algorithm is a universal attack vector, and this depends upon the input dependent attack used.

It is the simplicity of the algorithm which we would like to emphasize. The algorithm samples a set of $n$ images from the test set. We use the terms batch size and sample size interchangeably. For each of the sample points it computes an input-dependent attack direction. We stack these attack

---

[1] the authors consider other $\ell_p$ norms as well. We restrict ourselves to the $\ell_2$-norm.

directions in a matrix, one for each of the sample points and return the top singular vector of the matrix of attack directions as our SVD-Universal.

---

**Algorithm 1:** SVD-Universal Algorithm

---

**Data:** Network $N$, any input-dependent adversarial attack $A$, and $n$ test samples.
**Result:** A universal attack direction for network $N$
1  Obtain input-dependent perturbation vectors $a_1, ..., a_n$ based on $A$ for the $n$ test samples.
2  Normalize $a_i$'s to get only the attack directions or unit vectors $u_i = a_i/||a_i||_2$, for $i = 1$ to $n$.
3  Form a matrix $M$ whose rows are $u_1, u_2, \ldots, u_n$.
4  Compute Singular Value Decomposition (SVD) of $M$ as $M = USV^T$, with
    $V = [v_1|v_2|\ldots|v_n]$.
5  Return the top right singular vector $v_1$ as the universal attack vector.

---

### 4.2    ERROR RATES AND FOOLING RATES OF SVD-UNIVERSAL

We evaluate the fooling rate of SVD-Gradient, SVD-FGSM and SVD-DeepFool on the validation set of the ImageNet dataset on VGG16, VGG19 and ResNet50. We used off the shelf code available for these architectures. In these architectures pixel intensities of images are scaled down and images are normalized before they are deployed for use in classifiers. Our (unit) attack vectors are constructed using batch sizes of 64(0.13%), 128(0.26%)) and 1024(2%).

In Figure 1, we plot the fooling rate of $\epsilon \mathbf{v}$ as a function of $\epsilon$, in all the architectures, where $\mathbf{v}$ is obtained using a batch size of 64. The top left plot is the fooling rate of SVD-gradient, the top right that of SVD-FGSM and the bottom left that of SVD-DeepFool. We explain the bottom right plot in the next section. SVD-Gradient attack scaled to have norm 50 fools about 32% of the validation set of ImageNet on VGG19. Note that the average $\ell_2$ norm of this validation set is $450^2$ and using the language of Moosavi-Dezfooli et al. (2017a), a perturbation with norm 50 is also quasi-imperceptible. We visualize the perturbed images in Figure 4 in Appendix A.1.

We plot the error rate of SVD-Universal for ImageNet on VGG19 in Figure 5, Appendix A.1, on different batch sizes. It is clear from the plots that SVD-Universal attacks obtained using batch size of 64 perform as well (and sometimes better) as attack vectors obtained using larger batch sizes.

In Appendix A.2 we plot the error rate of SVD-Universal attacks on CIFAR-10 trained on ResNet18. On ResNet18, the error rate of SVD-DeepFool obtained from 100 samples of the CIFAR-10 test data is 0.55 when it is scaled to norm 15 (0.23 of the average norm), see Figure 6. We notice that the SVD-Universal perturbations constructed with small batch sizes perform better.

The visualizations of our perturbations as images are given in Appendix B.

### 4.3    COMPARISON WITH MOOSAVI-DEZFOOLI ET AL. (2017A), KHRULKOV & OSELEDETS (2018)

In this section we compare our results with that reported in Moosavi-Dezfooli et al. (2017a) and Khrulkov & Oseledets (2018) on the validation set of Imagenet. The pixel intensities of the dataset used in the experiments of Moosavi-Dezfooli et al. (2017a) and Khrulkov & Oseledets (2018) are in the range $[0, 255]$. The average $\ell_2$ norm of the validation set is 50,000 and the average $\ell_\infty$ norm is 250, Moosavi-Dezfooli et al. (2017a, Footnote, Page 4). The $\ell_2$ norm of the universal perturbation used in Moosavi-Dezfooli et al. (2017a) is 1/25 of 50,000 = 2000. The $\ell_\infty$ norm of the universal perturbation used in Khrulkov & Oseledets (2018) is 1/25 of 250 = 10. The pixel intensities of the validation set of Imagenet used in our experiments is normalized and the average $\ell_2$ norm of this normalized dataset is 450.

In (Khrulkov & Oseledets, 2018, Figure 9) the authors report that the universal perturbation of Moosavi-Dezfooli et al. (2017a) constructed from a batch size 64 and having $\ell_\infty$ norm 10 has a fooling rate of **0.14** on VGG19. A comparable perturbation in our model has $\ell_2$ norm 1/25 of 450 = 18. Figure 1 shows that SVD-Gradient and SVD-DeepFool perturbation with norm 18 give a fooling rate of **0.13** on VGG19.

---

[2]https://github.com/pytorch/examples/tree/master/imagenet

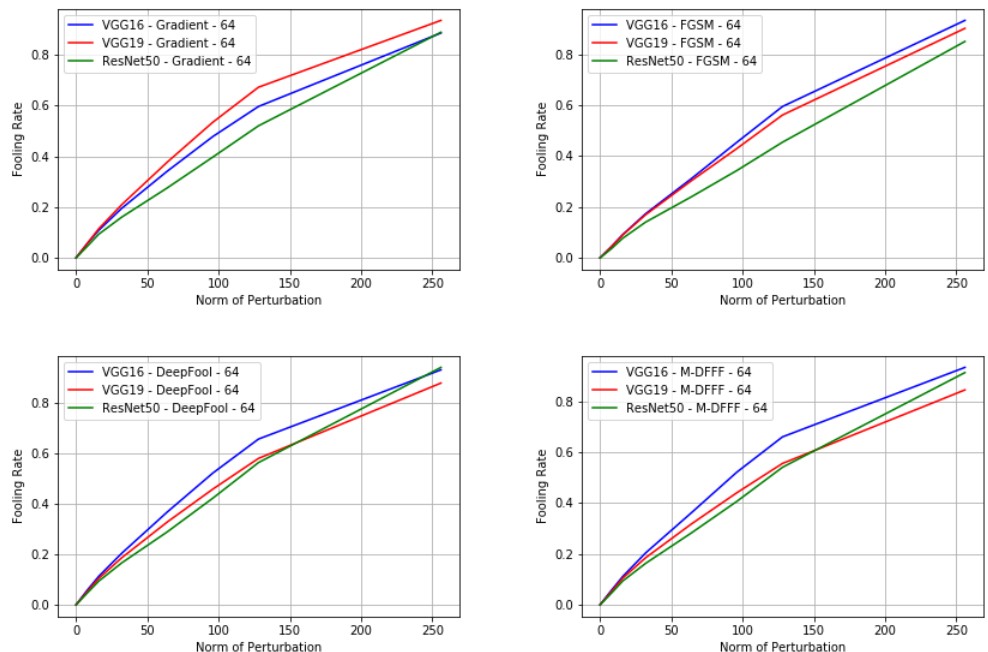

Figure 1: On ImageNet validation, VGG16 vs VGG19 vs ResNet50: fooling rate vs. norm of perturbation. Attacks constructed using 64 samples. (top left) *SVD-Gradient* (top right) *SVD-FGSM* (bottom left) *SVD-DeepFool* and (bottom right) M-DFFF universal.

To further compare our universal perturbations with that of Moosavi-Dezfooli et al. (2017a), the same 64 samples used to construct our SVD-Universal vectors were used to obtain the universal perturbation vector, M-DFFF, of Moosavi-Dezfooli et al. (2017a). This vector was scaled down to get a unit vector $w$ in $\ell_2$ norm. In Figure 1 (right bottom) we plot the fooling rate of M-DFFF $\epsilon w$ as a function of $\epsilon$, in all the architectures. We observe trends similar to what was reported in Moosavi-Dezfooli et al. (2017a, Table 1) - the error rate achieved by M-DFFF on VGG16 is higher than that of M-DFFF on VGG19. Observe from this figure that on VGG19, the M-DFFF universal perturbation of $\ell_2$ norm 18 gives a fooling rate of **0.13**.

The plots in Figure 1 of the fooling rates of the four universal attacks on VGG16 and ResNet50 are are collated together in Figure 2. We also plot the error rate of these universal attacks on VGG16 and ResNet50. SVD-DeepFool and M-DFFF have comparable fooling and error rates for ImageNet on the two networks.

Khrulkov & Oseledets (2018) get a fooling rate of more than **0.4** using batch size of 64 and $\ell_\infty$ norm 10. At this norm their universal attack is stronger than both our attack and that of Moosavi-Dezfooli et al. (2017a). However Khrulkov & Oseledets (2018) do an extensive experimentation and determine which intermediate layer to attack and $(p,q)$ are also optimized to maximize the fooling rate of their $(p,q)$-singular vector. However $(p,q)$-SVD computation is expensive and is known to be a hard problem, (Bhaskara & Vijayaraghavan, 2011; Bhattiprolu et al., 2019). We do no such optimization and use $p = q = 2$, our emphasis being on the simplicity of our SVD-Universal algorithm. We also observe a trend similar to what is reported by Khrulkov & Oseledets (2018) - the error rate of SVD-Universal attacks is higher on VGG16 and VGG19 than on ResNet50.

Figure 7 in Appendix A.2 compares the error rates of SVD-Universal attacks and M-DFFF constructed using samples of size 100 from the test data of CIFAR-10 on ResNet18. Error rates of SVD-DeepFool and M-DFFF are comparable - for this dataset, SVD-Gradient is not as good. When the norm of the perturbation is about 15 (0.23 of the average norm), the error rate of SVD-DeepFool fool is about 0.55.

**VGG16**                                                   **ResNet50**

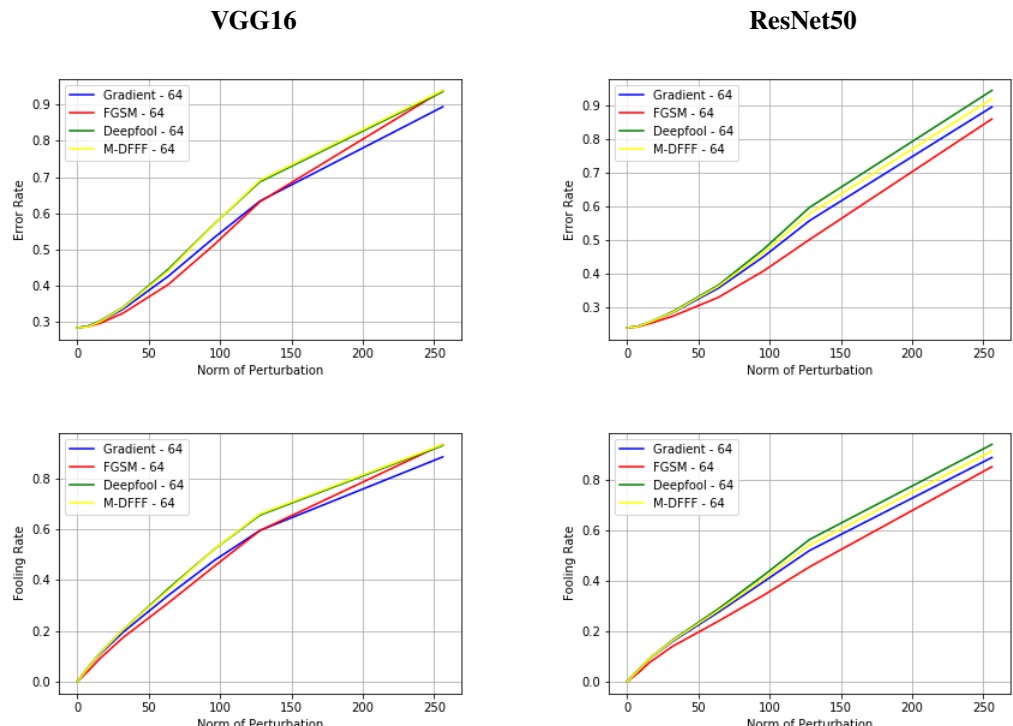

Figure 2: On ImageNet validation, (top left): VGG16: error rate (bottom left) VGG16: fooling rate (top right) ResNet50: error rate, (bottom right) ResNet50: fooling rate, vs. norm of perturbation along top singular vector of attack directions on 64 samples.

## 5 THEORETICAL ANALYSIS OF OUR UNIVERSAL PERTURBATION

In this section, we attempt to provide a theoretical justification for the existence of universal adversarial perturbations. Let our data be given by a joint distribution $\mathcal{D}$ on feature vectors in $\mathbb{R}^d$ and corresponding true labels in $[k] = \{1, 2, \ldots, k\}$. Let $(X, Y)$ denote a random sample from $\mathcal{D}$. Let $f : \mathbb{R}^d \to [k]$ be a given classifier, and for any $\mathbf{x} \in \mathbb{R}^d$, let $\mathcal{A}(\mathbf{x})$ be its adversarial perturbation given by a fixed adversarial attack $\mathcal{A}$, e.g., *FGSM*, *DeepFool*. As in Moosavi-Dezfooli et al. (2017b), the fooling rate of this attack is given by $\Pr\left(f(X) \neq f(X + \mathcal{A}(X))\right)$, where the probability is over the data distribution $\mathcal{D}$.

Define $A = \{\mathbf{x} : f(\mathbf{x}) \neq f(\mathbf{x} + \mathcal{A}(\mathbf{x}))\}$. For any $\mathbf{x} \in A$, assume that $\mathbf{x} + \mathcal{A}(\mathbf{x})$ lies on the decision boundary, and let the hyperplane $H_{\mathbf{x}} = \{\mathbf{x} + \mathbf{z} \in \mathbb{R}^d : \langle \mathbf{z}, \mathcal{A}(\mathbf{x}) \rangle = \|\mathcal{A}(\mathbf{x})\|_2^2\}$ be a local, linear approximation to the decision boundary at $\mathbf{x} + \mathcal{A}(\mathbf{x})$. This holds for adversarial attacks such as DeepFool by Moosavi-Dezfooli et al. (2016) which try to find an adversarial perturbation $\mathcal{A}(\mathbf{x})$ such that $\mathbf{x} + \mathcal{A}(\mathbf{x})$ is the nearest point to $\mathbf{x}$ on the decision boundary. Now consider the halfspace $S_{\mathbf{x}} = \{\mathbf{x} + \mathbf{z} \in \mathbb{R}^d : \langle \mathbf{z}, \mathcal{A}(\mathbf{x}) \rangle \geq \|\mathcal{A}(\mathbf{x})\|_2^2\}$. Note that $\mathbf{x} \notin S_{\mathbf{x}}$ and $\mathbf{x} + \mathcal{A}(\mathbf{x}) \in S_{\mathbf{x}}$. *For simplicity of analysis, we assume that $f(\mathbf{x}) \neq f(\mathbf{x} + \mathbf{z})$, for all $\mathbf{x} \in A$ and $\mathbf{x} + \mathbf{z} \in S_{\mathbf{x}}$.* This is a reasonable assumption in a small neighborhood of $\mathbf{x}$, and is implied by the positive curvature of the decision boundary assumed in the analysis of Moosavi-Dezfooli et al. (2017b). In other words, we assume that if an adversarial perturbation such as $\mathcal{A}(\mathbf{x})$ fools the model at input $\mathbf{x}$, then any perturbation $\mathbf{z}$, where $\mathbf{z}$ has a sufficient projection along $\mathcal{A}(\mathbf{x})$, also fools the model at $\mathbf{x}$, and this is a reasonable assumption in a small neighborhood of $\mathbf{x}$.

**Theorem 1.** *Given any joint data distribution $\mathcal{D}$ on features or inputs in $\mathbb{R}^d$ and true labels in $[k]$, let $(X, Y)$ denote a random sample from $\mathcal{D}$. For any $\mathbf{x} \in \mathbb{R}^d$, let $\mathcal{A}(\mathbf{x})$ be its adversarial perturbation by a fixed input-dependent adversarial attack $\mathcal{A}$. Let*

$$M = \mathbb{E}\left[\frac{\mathcal{A}(X)}{\|\mathcal{A}(X)\|_2}\frac{\mathcal{A}(X)^T}{\|\mathcal{A}(X)\|_2}\right] \in \mathbb{R}^{d \times d},$$

and $0 \leq \lambda \leq 1$ be the top eigenvalue of $M$ and $\mathbf{v} \in \mathbb{R}^d$ be the corresponding eigenvector (normalized to have unit $\ell_2$ norm). Then under the assumption that $f(\mathbf{x}) \neq f(\mathbf{x} + \mathbf{z})$, for all $\mathbf{x} \in A$ and $\mathbf{x} + \mathbf{z} \in S_{\mathbf{x}}$, we have

$$\Pr\left(f(X + \mathbf{u}) \neq f(X)\right) \geq \Pr\left(f(X) \neq f(X + \mathcal{A}(X))\right) - \frac{1 - \lambda}{1 - \delta^2},$$

where $\mathbf{u} = \pm(\epsilon/\delta)\mathbf{v}$, where $\epsilon = \max_{\mathbf{x}} \|\mathcal{A}(\mathbf{x})\|_2$.

*Proof.* Let $\mu(\mathbf{x})$ denote the induced probability density on features or inputs by the distribution $\mathcal{D}$. Define $A = \{\mathbf{x} : f(\mathbf{x}) \neq f(\mathbf{x} + \mathcal{A}(\mathbf{x}))\}$ and $G = \{\mathbf{x} : |\langle \mathcal{A}(\mathbf{x}), \mathbf{v}\rangle| \geq \delta \|\mathcal{A}(\mathbf{x})\|_2\}$. Since $\lambda$ is the top eigenvalue of $M$ with $\mathbf{v}$ as its corresponding (unit) eigenvector,

$$\begin{aligned}
\lambda &= \mathbb{E}\left[\left\langle \frac{\mathcal{A}(X)}{\|\mathcal{A}(X)\|_2}, \mathbf{v}\right\rangle^2\right] \\
&= \int_{\mathbf{x} \in G} \left\langle \frac{\mathcal{A}(\mathbf{x})}{\|\mathcal{A}(\mathbf{x})\|_2}, \mathbf{v}\right\rangle^2 \mu(\mathbf{x})d\mathbf{x} + \int_{\mathbf{x} \notin G} \left\langle \frac{\mathcal{A}(\mathbf{x})}{\|\mathcal{A}(\mathbf{x})\|_2}, \mathbf{v}\right\rangle^2 \mu(\mathbf{x})d\mathbf{x} \\
&\leq \int_{\mathbf{x} \in G} \mu(\mathbf{x})d\mathbf{x} + \delta^2 \int_{\mathbf{x} \notin G} \mu(\mathbf{x})d\mathbf{x} \quad \text{because } \|\mathbf{v}\|_2 = 1 \\
&= \Pr(G) + \delta^2(1 - \Pr(G)) \\
&= (1 - \delta^2)\Pr(G) + \delta^2.
\end{aligned}$$

Thus, $\Pr(G) \geq (\lambda - \delta^2)/(1 - \delta^2)$, and equivalently, $\Pr(G^c) = 1 - \Pr(G) \leq (1 - \lambda)/(1 - \delta^2)$. Now for any $\mathbf{x} \in G$, we have $|\langle \mathcal{A}(\mathbf{x}), \mathbf{v}\rangle| \geq \delta \|\mathcal{A}(\mathbf{x})\|_2$. Letting $\epsilon = \max_{\mathbf{x}} \|\mathcal{A}(\mathbf{x})\|_2$, we get $|\langle \mathcal{A}(\mathbf{x}), (\epsilon/\delta)\mathbf{v}\rangle| \geq \|\mathcal{A}(\mathbf{x})\|_2^2$. Thus, $\mathbf{x} \pm (\epsilon/\delta)\mathbf{v} \in S_{\mathbf{x}}$, where $S_{\mathbf{x}} = \{\mathbf{x} + \mathbf{z} \in \mathbb{R}^d : \langle \mathbf{z}, \mathcal{A}(\mathbf{x})\rangle \geq \|\mathcal{A}(\mathbf{x})\|_2^2\}$, and therefore, by our assumption stated before Theorem 1, we have $f(\mathbf{x}) \neq f(\mathbf{x} + \mathbf{u})$, where $\mathbf{u} = \pm(\epsilon/\delta)\mathbf{v}$. Putting all of this together, $\Pr(f(X + \mathbf{u}) \neq f(X)) \geq \Pr(G \cap A) \geq \Pr(A) - \Pr(A \cap G^c) \geq \Pr(A) - \Pr(G^c)$, and therefore,

$$\Pr\left(f(X + \mathbf{u}) \neq f(X)\right) \geq \Pr\left(f(X) \neq f(X + \mathcal{A}(X))\right) - \frac{1 - \lambda}{1 - \delta^2},$$

□

Theorem 1 shows that any norm-bounded, input-dependent, adversarial attack $\mathcal{A}$ can be converted into a universal attack $\mathbf{u}$ of comparable norm, without losing much in the fooling, if the top eigenvalue $\lambda$ of $M$ is close to 1. This universal attack direction lies in the one-dimensional span of the top eigenvector $\mathbf{v}$ of $M$. The proof of Theorem 1 can be easily generalized to the top SVD subspace of $M$ and where the top few eigenvalues of $M$ dominate its spectrum (note that $\text{tr}(M) = 1$).

**Singular value drop** We empirically verify our hypothesis about top eigenvalue (or the top few eigenvalues) dominating the spectrum of $M$ in Theorem 1. Let $X_1, X_2, \ldots, X_m$ be $m$ i.i.d. samples of $X$ drawn from the distribution $\mathcal{D}$ and consider the unnormalized, empirical analog of $M$ as follows,

$$\sum_{i=1}^m \frac{\mathcal{A}(X_i)}{\|\mathcal{A}(X_i)\|_2} \frac{\mathcal{A}(X_i)^T}{\|\mathcal{A}(X_i)\|_2}.$$

Figure (3) shows how the singular values drop for the three input dependent attacks, *Gradient*, *FGSM*, and *DeepFool* on CIFAR-10 trained on ResNet18 on batch sizes 500 and 10,000. These plots indicate that the drop in singular values is a shared phenomenon across different input dependent attacks, and the trend is similar even when we look at a small number of input samples.

Our second contribution is finding a good approximation to the *universal* adversarial perturbation given by the top eigenvector $\mathbf{v}$ of $M$, using only a small sample $X_1, X_2, \ldots, X_m$ from the distribution $\mathcal{D}$. Theorem 2 shows that we can efficiently pick such a small sample whose size is independent of $\mathcal{D}$, depends linearly on the *intrinsic dimension* of $M$, and logarithmically on the feature dimension $d$.

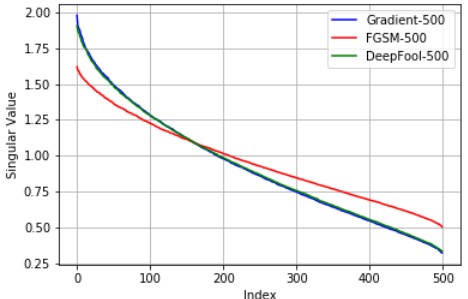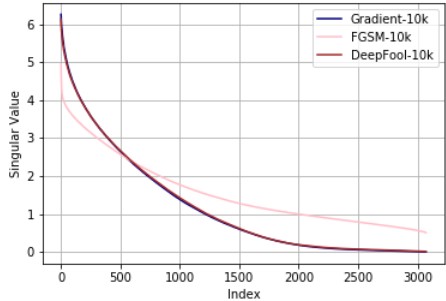

Figure 3: On CIFAR-10, ResNet18, Singular values of attack directions over a sample of (left) 500 and (right) 10,000 test points.

**Theorem 2.** *Given any joint data distribution $\mathcal{D}$ on features in $\mathbb{R}^d$ and true labels in $[k]$, let $(X, Y)$ denote a random sample from $\mathcal{D}$. For any $\mathbf{x} \in \mathbb{R}^d$, let $\mathcal{A}(\mathbf{x})$ denote the adversarial perturbation of $\mathbf{x}$ according a fixed input-dependent adversarial attack $\mathcal{A}$. Let*

$$M = \mathbb{E}\left[\frac{\mathcal{A}(X)}{\|\mathcal{A}(X)\|_2} \frac{\mathcal{A}(X)^T}{\|\mathcal{A}(X)\|_2}\right] \in \mathbb{R}^{d \times d}.$$

*Let $0 \leq \lambda = \|M\|_2 \leq 1$ denote the top eigenvalue of $M$ and let $v$ denote its corresponding eigenvector (normalized to have unit $\ell_2$ norm). Let $r = \operatorname{tr}(M) / \|M\|_2$ be the* intrinsic dimension *of $M$. Let $X_1, X_2, \ldots, X_m$ be $m$ i.i.d. samples of $X$ drawn from the distribution $\mathcal{D}$, and let $\tilde{\lambda} = \left\|\tilde{M}\right\|_2$ be the top eigenvalue of the matrix $\tilde{M}$,*

$$\tilde{M} = \frac{1}{m} \sum_{i=1}^m \frac{\mathcal{A}(X_i)}{\|\mathcal{A}(X_i)\|_2} \frac{\mathcal{A}(X_i)^T}{\|\mathcal{A}(X_i)\|_2},$$

*and $\tilde{\mathbf{v}}$ be the top eigenvector of $\tilde{M}$.*

*Also suppose that there is a gap of at least $\gamma\lambda$ between the top eigenvalue $\lambda$ and the second eigenvalue of $M$. Then for any $0 \leq \epsilon < \gamma$ and $m = O(\epsilon^{-2} r \log d)$, we get $\|\mathbf{v} - \tilde{\mathbf{v}}\|_2 \leq \epsilon/\gamma$, with a constant probability. This probability can be boosted to $1 - \delta$ by having an additional $\log(1/\delta)$ in the $O(\cdot)$.*

*Proof.* Take $m = O(\epsilon^{-2} r \log d)$. By the covariance estimation bound in Theorem 4 and Markov's inequality, we get that $\left\|M - \tilde{M}\right\|_2 \leq \epsilon\lambda$, with a constant probability. Applying Theorem 5 due to Weyl on eigenvalue perturbation, we get $\left|\lambda - \tilde{\lambda}\right| \leq \epsilon\lambda$. Moreover, if there is gap of at least $\gamma\lambda$ between the first and the second eigenvalue of $M$ with $\gamma > \epsilon$, we can use the well-known Theorem 6 of Davis-Kahan to bound the difference between the corresponding eigenvectors as $\|\mathbf{v} - \tilde{\mathbf{v}}\|_2 \leq \epsilon/\gamma$, with a constant probability. Please see Appendix D, and the book by Vershynin (2018) cited therein, for more details about the covariance estimation bound, Weyl's theorem, and Davis-Kahan theorem. $\square$

The theoretical bounds are weaker than our empirical observations about the number of test samples needed for the universal attack. What we want to highlight is that the bound in Theorem 2 is independent of the support of underlying data distribution $\mathcal{D}$ and depends only on its dimensionality $d$. There is more room to tighten our analysis using additional properties of the data distribution as well as the spectral properties of $M$.

## 6    UNIVERSAL INVARIANT PERTURBATIONS AND FUTURE DIRECTIONS

Let $\mathbf{x}$ be an image for which the predicted label of a neural network classifier $f$ is $f(\mathbf{x})$. For highly accurate classifiers trained on data that already contains small rotation augmentations, one would

expect $f(\mathbf{x} + \Delta(\mathbf{x})) = f(\mathbf{x})$, where $\mathbf{x} + \Delta(\mathbf{x})$ is the image resulting from a small rotation applied to $\mathbf{x}$. For an image $\mathbf{x}$, we call $\mathcal{I}(\mathbf{x})$ an invariant perturbation for $\mathbf{x}$, if $f(\mathbf{x} + \mathcal{I}(\mathbf{x})) = f(\mathbf{x})$. Using this terminology we expect $\Delta(\mathbf{x})$ to be an invariant perturbation for $\mathbf{x}$ for classifiers trained on data that is inherently (or additionally) augmented with small rotations. A natural question is whether there are any universal invariant perturbations $\mathbf{x} + \mathbf{u}$ that are invariant for most $\mathbf{x}$. We show empirically that there are such universal invariant perturbations. By their very definition we would also expect that universal invariant perturbations will not be universal adversarial perturbations. We show this empirically by plotting the error rates of the top singular invariant perturbation of CIFAR-10 on ResNet18 in Figure 10, Appendix C. The error rate is below 0.2 even when the norm of the invariant perturbation is scaled up to 50. We give more empirical evidence of such invariant perturbations - we show that the principal angles between the subspace of the top 5 universal adversarial perturbations and the subspace of the top 5 universal invariant perturbations are all close to $90°$.

The empirical and theoretical results in this paper hold in greater generality than what we have presented. Our experiments indicate the existence of universal adversarial perturbations and universal invariant perturbations for datasets trained on equivariant networks. GCNN's Cohen & Welling (2016) are among the more popular equivariant networks, and achieve state of the art on CIFAR-10. In Appendix C we give a peek into this work and show plots of the error rates of SVD-Universal on CIFAR-10 on GCNN's. We also plot the singular values of CIFAR-10 on GCNN's.

It is common to augment data with rotations and train neural networks, so they become equivariant to rotations and so an image and its rotation are classified the same. We believe that a study of how the subspaces of universal adversarial perturbations and universal invariant perturbations change as a model is trained to handle more and more rotations will be important. This is ongoing work.

## 7 CONCLUSION

We show how to use a small sample of input-dependent adversarial attack directions on test inputs to find a single universal adversarial perturbation that results in a large drop in the accuracy of state of the art neural network models. Our main observation is a spectral property shared by different attacks directions such as *Gradients*, *FGSM*, *DeepFool*. We give a theoretical justification for how this spectral property helps in using the top singular vector as a universal attack direction. We justify theoretically and empirically that such a perturbation can be computed using only a small sample of test inputs. Our empirical results also indicate that these universal perturbations are almost orthogonal to perturbations of inputs that these models are invariant to.

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

## APPENDIX A    DATA SETS AND MODEL ARCHITECTURES USED

**Data sets**    CIFAR-10 dataset consists of $60,000$ images of $32 \times 32$ size, divided into 10 classes. $40,000$ used for training, $10,000$ for validation and $10,000$ for testing. ImageNet dataset refers to ILSRVC 2012 dataset (Russakovsky et al., 2015) which consists of images of $224 \times 224$ size, divided into 1000 classes. We use the validation set of $50,000$ images for our testing purposes.

All experiments performed on neural network-based models were done using the validation set of ImageNet and test set of CIFAR-10 datasets.

**Model Architectures**    For the ImageNet based experiments we use pre-trained networks of VGG16, VGG19 and ResNet50 architectures[3]. For the CIFAR-10 based experiments we use the ResNet18 architecture as in He et al. (2016).

---

[3]https://pytorch.org/docs/stable/torchvision/models.html

## A.1 SVD-UNIVERSAL ON IMAGENET

Figure 4 shows that the images with SVD-DeepFool perturbations upto norm 100 are quasi-imperceptible. Figure 5 shows the error rate of SVD-Universal on VGG19 calculated with varying sample size.

| Norm 0 | Norm 16 | Norm 50 | Norm 100 | Norm 450 |
|--------|---------|---------|----------|----------|

Figure 4: Sample images misclassified at norm 16. Images perturbed with SVD-DeepFool.

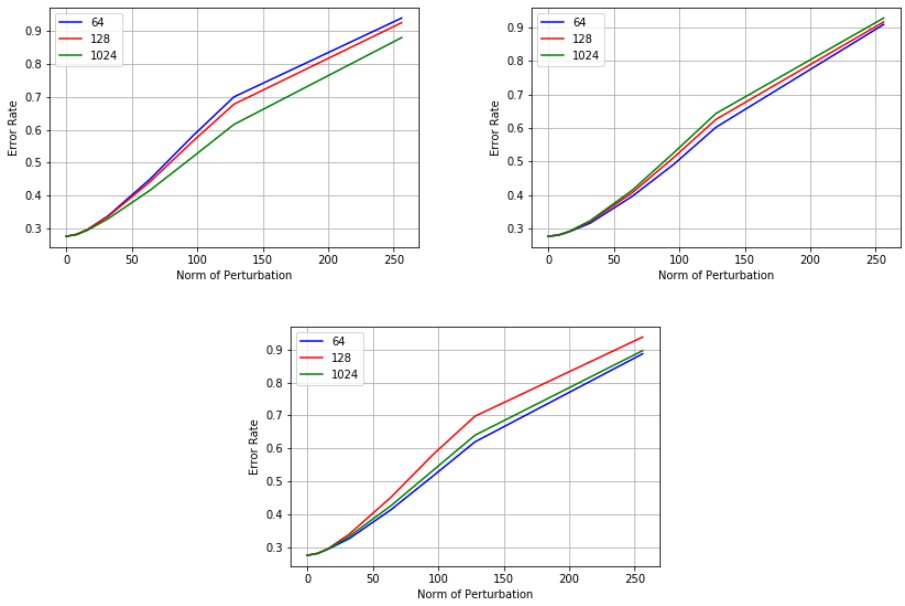

Figure 5: On ImageNet validation, error rate, VGG19: error rate vs. norm of perturbation along top singular vector of attack directions on 64/128/1024 sample, (top left) *Gradient* (top right) *FGSM* (bottom) *DeepFool*

## A.2  SVD-UNIVERSAL ON CIFAR-10

In this section we plot the error rates of SVD-Universal on CIFAR-10 trained on ResNet18.

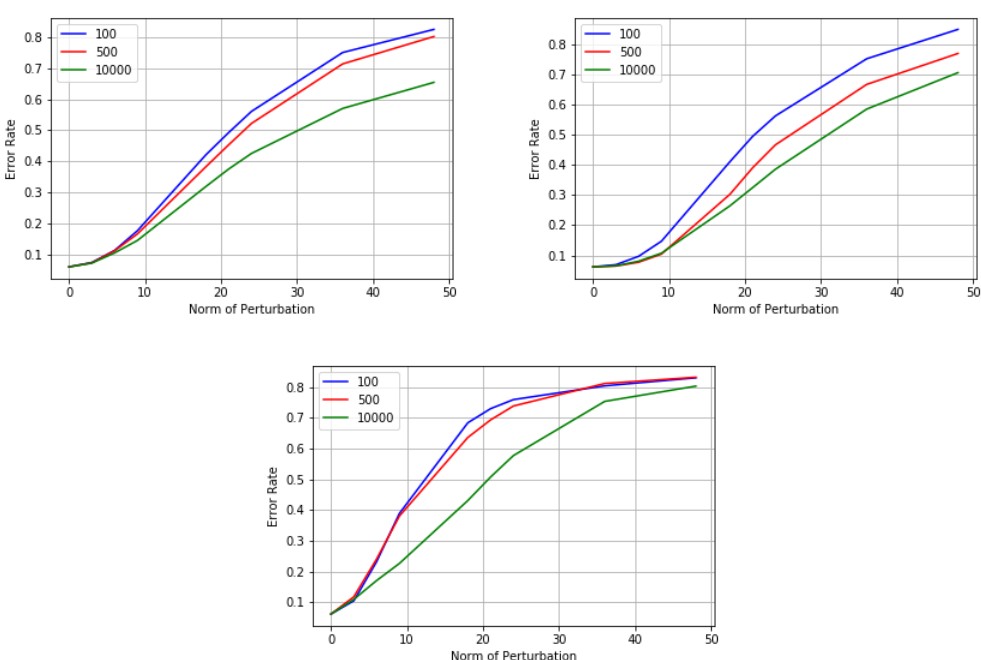

Figure 6: On CIFAR-10, ResNet18: error rate vs. norm of perturbation along top singular vector of attack directions on 100/500/10000 sample, (top left) *Gradient* (top right) *FGSM* (bottom) *DeepFool*

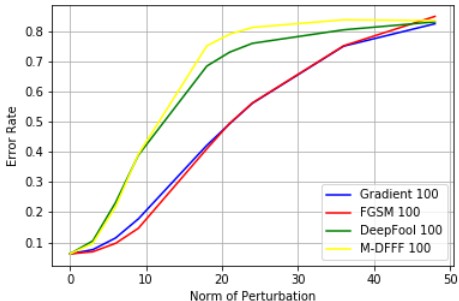

Figure 7: On CIFAR-10, ResNet18: error rate vs. norm of perturbation along top singular vector of attack directions on 100 samples.

## APPENDIX B    VISUALIZING SVD-UNIVERSAL PERTURBATIONS

We visualize the top singular vectors for the *Gradient*, *FGSM*, *DeepFool* directions from ImageNet and CIFAR-10 in this section.



Figure 8: For ImageNet, Top SVD vector from (left) Gradient, (center) FGSM, (right) DeepFool on ResNet50.

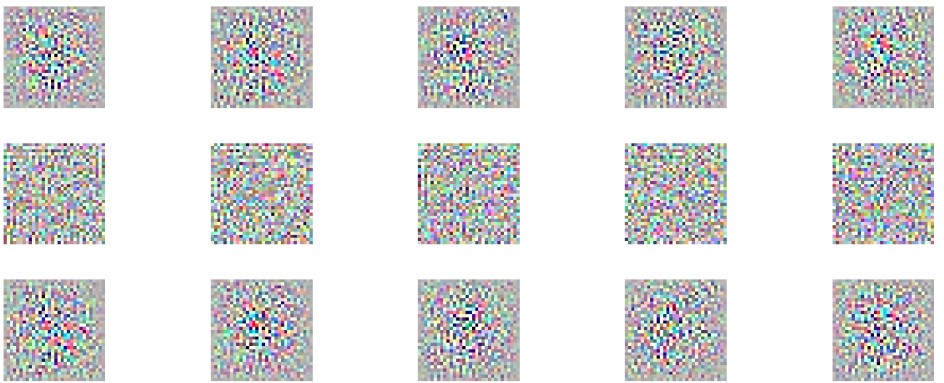

Figure 9: For CIFAR-10 (top) Top 5 SVD vectors from *Gradient*, (middle) Top 5 SVD vectors from *FGSM*, (bottom) Top 5 SVD vectors from DeepFool on ResNet18.

## APPENDIX C    INVARIANT PERTURBATIONS

### C.1    PRINCIPAL COMPONENTS OF ATTACK AND INVARIANT DIRECTIONS

To construct the matrix, we pick a batch of 100 images, and for each image x in our sample we compute the difference between x and its 2 degree rotation. We stack these (input dependent invariant directions) as rows of the matrix and compute the top singular (unit) vector $\mathbf{w}$. We call $\mathbf{w}$ the universal invariant perturbation. Figure 10 shows the error rate of $\epsilon\mathbf{w}$ as a function of $\epsilon$ on CIFAR-10. The curves are flat indicating that these directions are invariant perturbations.

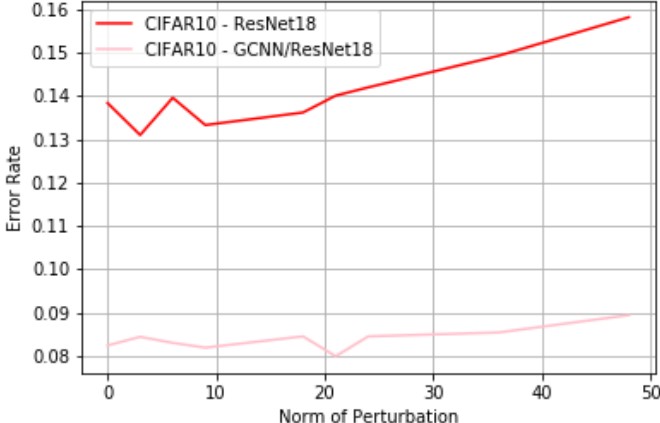

Figure 10: Error rate vs norm of perturbation using top singular invariant vector for StdCNN and GCNN on CIFAR-10 test data.

Recall that for two subspaces $V,W$, the first principal angle is defined as the minimum angle between two unit vectors $v_1 \in V$, $w_1 \in W$. The second principal angle is the minimum angle between unit vectors $v_2 \in V, w_2 \in W$, with $v_2 \perp v_1$ and $w_2 \perp w_1$. The other principal angles are defined similarly.

Table 1: Principal angles between Top-5 SVD-subspace of gradient directions of test points, and the Top-5 SVD-subspace of invariant directions on test points (small $2°$ rotations) respectively.

| Dataset | Model | 1 | 2 | 3 | 4 | 5 |
|---------|-------|---|---|---|---|---|
| CIFAR-10 | ResNet18 | 89.99958413 | 89.82902968 | 89.52802116 | 88.91712082 | 88.64889062 |

## APPENDIX D  PROOF DETAILS

Theorem 5.6.1 (General covariance estimation) from Vershynin (2018) bounds the spectral norm of covariance matrix estimated from a small number of samples as follows.

**Theorem 3.** *Let $X$ be a random vector in $\mathbb{R}^d$, $d \geq 2$. Assume that for some $K \geq 1$, $\|X\|_2 \leq K \left( \mathbb{E} \left[ \|X\|_2^2 \right] \right)^{1/2}$, almost surely. Let $\Sigma = \mathbb{E} \left[ X X^T \right]$ be the covariance matrix of $X$ and $\Sigma_m = \frac{1}{m} \sum_{i=1}^{m} X_i X_i^T$ be the estimated covariance from $m$ i.i.d. samples $X_1, X_2, \ldots, X_m$. Then for every positive integer $m$, we have*

$$\mathbb{E} \left[ \|\Sigma_m - \Sigma\|_2 \right] \leq C \left( \sqrt{\frac{K^2 d \log d}{m}} + \frac{K^2 d \log d}{m} \right) \|\Sigma\|_2 ,$$

*for some positive constant $C$ and $\|\Sigma\|_2$ being the spectral norm (or the top eigenvalue) of $\Sigma$.*

Note that using $m = O(\epsilon^{-2} d \log d)$ we get $\mathbb{E} \left[ \|\Sigma_m - \Sigma\|_2 \right] \leq \epsilon \|\Sigma\|_2$.

A tighter version of Theorem 5.6.1 appears as Remark 5.6.3, when the *intrinsic dimension* $r = \text{tr}(\Sigma) / \|\Sigma\|_2 \ll d$.

**Theorem 4.** *Let $X$ be a random vector in $\mathbb{R}^d$, and $d \geq 2$. Assume that for some $K \geq 1$, $\|X\|_2 \leq K \left( \mathbb{E} \left[ \|X\|_2^2 \right] \right)^{1/2}$, almost surely. Let $\Sigma = \mathbb{E} \left[ X X^T \right]$ be the covariance matrix of $X$ and $\Sigma_m = \frac{1}{m} \sum_{i=1}^{m} X_i X_i^T$ be the estimated covariance from $m$ i.i.d. samples $X_1, X_2, \ldots, X_m$. Then for every positive integer $m$, we have*

$$\mathbb{E} \left[ \|\Sigma_m - \Sigma\|_2 \right] \leq C \left( \sqrt{\frac{K^2 r \log d}{m}} + \frac{K^2 r \log d}{m} \right) \|\Sigma\|_2 ,$$

*for some positive constant $C$ and $\|\Sigma\|_2$ being the operator norm (or the top eigenvalue) of $\Sigma$.*

Note that using $m = O(\epsilon^{-2} r \log d)$ we get $\mathbb{E} \left[ \|\Sigma_m - \Sigma\|_2 \right] \leq \epsilon \|\Sigma\|_2$.

Theorem 4.5.3 (Weyl's Inequality) from Vershynin (2018) upper bounds the difference between $i$-th eigenvalues of two symmetric matrices $A$ and $B$ using the spectral norm of $A - B$.

**Theorem 5.** *For any two symmetric matrices $A$ and $B$ in $\mathbb{R}^{d \times d}$, $|\lambda_i(A) - \lambda_i(B)| \leq \|A - B\|_2$, where $\lambda_i(A)$ and $\lambda_i(B)$ are the $i$-th eigenvalues of $A$ and $B$, respectively.*

In other words, the spectral norm of matrix perturbation bounds the stability of its spectrum.

Here is a special case of Theorem 4.5.5 (Davis-Kahan Theorem) and its immediate corollary mentioned in Vershynin (2018).

**Theorem 6.** *Let $A$ and $B$ be symmetric matrices in $\mathbb{R}^{d \times d}$. Fix $i \in [d]$ and assume that the largest eigenvalue of $A$ is well-separated from the rest of the spectrum, that is, $\lambda_1(A) - \lambda_2(A) \geq \delta > 0$. Then the angle $\theta$ between the top eigenvectors $v_1(A)$ and $v_1(B)$ of $A$ and $B$, respectively, satisfies $\sin \theta \leq 2 \|A - B\|_2 / \delta$.*

As an easy corollary, it implies that the top eigenvectors $v_1(A)$ and $v_1(B)$ are close to each other up to a sign, namely, there exists $s \in \{-1, 1\}$ such that

$$\|v_1(A) - s \, v_1(B)\|_2 \leq \frac{2^{3/2} \|A - B\|_2}{\delta}.$$

## APPENDIX E  ERROR RATES OF CIFAR-10 ON GCNN'S

For a fair comparison, the GCNN configuration used for CIFAR-10 is the same as that of ResNet18, except that the operations going from one layer to the next are replaced by equivalent GCNN operations, as given in Cohen & Welling (2016).

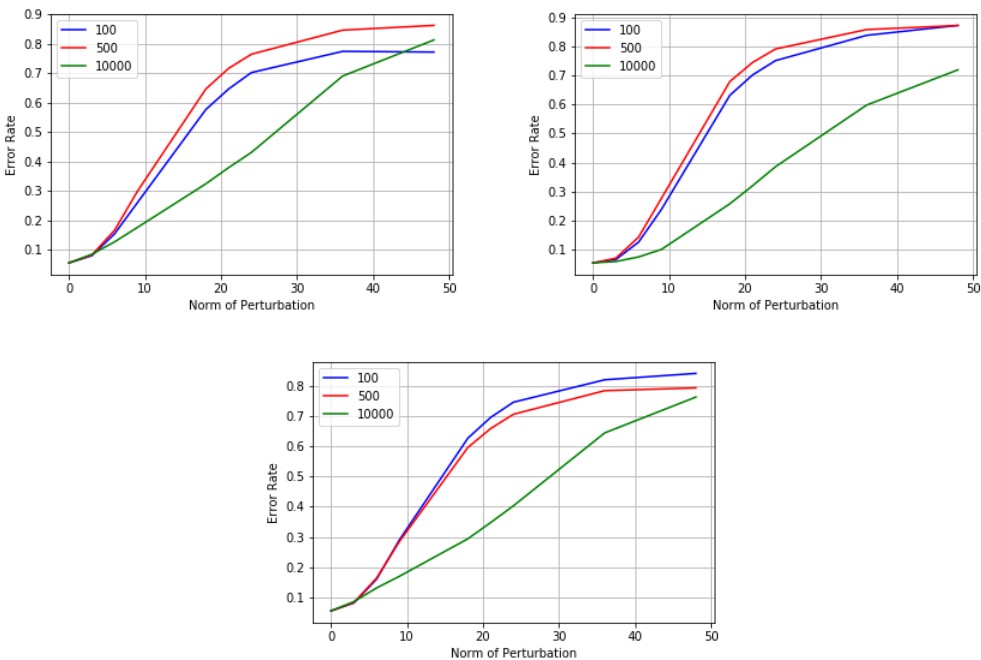

Figure 11: On CIFAR-10, GCNN: error rate vs. norm of perturbation along top singular vector of attack directions on 100/500/10000 samples, (top left) Gradient (top right) FGSM (bottom) Deep-Fool

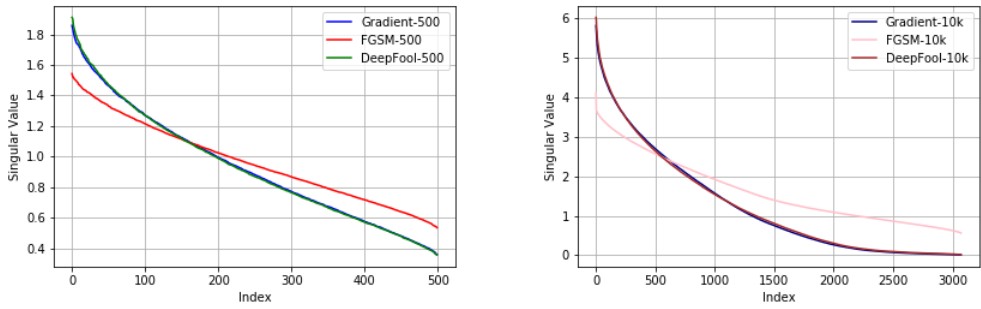

Figure 12: On CIFAR-10, GCNN/ResNet18, Singular values of attack directions over a sample of (left) 500 and (right) 10,000 test points.

Table 2: Principal angles between Top-5 SVD-subspace of gradient directions of test points, and the Top-5 SVD-subspace of invariant directions on test points (small 2° rotations) respectively.

| Dataset | Model | 1 | 2 | 3 | 4 | 5 |
|---------|-------|---|---|---|---|---|
| CIFAR-10 | GCNN/ResNet18 | 89.90921558 | 89.82469625 | 89.59947099 | 89.31534574 | 88.96739966 |

## APPENDIX F    TABLES REQUESTED BY REVIEWERS

Table 3: On ImageNet validation, VGG16 vs VGG19 vs ResNet50 vs M-DFFF: fooling rate vs. norm of perturbation in tabular form. Attacks constructed using 64 samples.

| Network | Vector (using 64 samples) | Norm 8 | Norm 16 | Norm 32 | Norm 64 |
|---|---|---|---|---|---|
| VGG16 | SVD-Gradient | 0.05720 | 0.10870 | 0.19470 | 0.34100 |
| VGG16 | SVD-FGSM | 0.04450 | 0.09182 | 0.17364 | 0.31124 |
| VGG16 | SVD-DeepFool | 0.05894 | 0.11334 | 0.20374 | 0.36744 |
| VGG16 | M-DFFF | 0.05720 | 0.11192 | 0.20426 | 0.36154 |
| VGG19 | SVD-Gradient | 0.05728 | 0.11490 | 0.20892 | 0.37672 |
| VGG19 | SVD-FGSM | 0.04366 | 0.09052 | 0.16956 | 0.30300 |
| VGG19 | SVD-DeepFool | 0.05376 | 0.10492 | 0.18602 | 0.32786 |
| VGG19 | M-DFFF | 0.05256 | 0.10542 | 0.18528 | 0.31764 |
| ResNet50 | SVD-Gradient | 0.04604 | 0.09334 | 0.16008 | 0.27532 |
| ResNet50 | SVD-FGSM | 0.03600 | 0.07740 | 0.14062 | 0.23984 |
| ResNet50 | SVD-DeepFool | 0.04742 | 0.09400 | 0.16580 | 0.28880 |
| ResNet50 | M-DFFF | 0.04602 | 0.09398 | 0.16318 | 0.28242 |

