# OpenReview forum: "Universal Adversarial Attack Using Very Few Test Examples"
_ICLR.cc/2020/Conference — Reject_

### Official Review · AnonReviewer1 · 2019-10-22
**Official Blind Review #1**

**Rating:** 3

**Review:**

This paper studied the problem of universal adversarial attack which is an input-agnostic perturbation. The authors proposed to use the top singular vector of input-dependent adversarial attack directions to perform universal adversarial attacks. The authors evaluated the error rates and fooling rates for three attacks on standard benchmark datasets.

- The paper is generally well-written and easy to follow. My main concern towards this paper is about the experiments part from several aspects. First, the proposed method needs quite large L2 norm (50 on ImageNet) to work, while common adversarial attack experiments on ImageNet are usually conducted with L2 perturbation strength of 5 or less. I totally understand that performing universal attack would be much more difficult, yet having such loose L2 norm constraint still seems impractical. Second, the authors did not compare with any other baselines such as  (Moosavi-Dezfooli et al. 2017a) arguing that their universal attack is different for different perturbation strength and pixels are normalized. I do not think normalized pixel will be a problem as you can simply scale the perturbation strength accordingly. And because (Moosavi-Dezfooli et al. 2017a) uses different attack vectors for different perturbation strength, some comparison between these two types of universal attacks should be presented in order to mark the difference and demonstrate your advantages. I would suggest the authors to compare with several mentioned baselines in the paper to show the superiority of the proposed method.

- Theorem 1 seems interesting, yet it needs a special assumption. The authors argue that this is a reasonable assumption in a small neighborhood of x. I wonder if the authors could conduct some demonstrative experiments to verify this? Because the definition of S_x depends on the attack function, does it mean that the assumption need to be held for any attack function? Also regarding the choice of \delta, it seems that \delta is different for different x? If so, since u is also depend on \delta, this attack vector seems not universal?


Detailed comments:
- In proof of Theorem 1, all S should be G?
- In proof of Theorem 2, how to get \|v - \hat v\|_2 \leq \epsilon/(\gamma - \epsilon)? Directly applying the Theorems seems to get \epsilon / (\gamma) only?

Depending on whether the authors can address my concerns, I may change the final rating.


======================
after the rebuttal

I thank the authors for their response but I still feel that the assumption is not well-justified and there is still a lot to improve in terms of experiments. Therefore I decided to keep my score unchanged.

**Experience Assessment:**

I have published in this field for several years.

**Review Assessment: Checking Correctness Of Derivations And Theory:**

I carefully checked the derivations and theory.

**Review Assessment: Checking Correctness Of Experiments:**

I assessed the sensibility of the experiments.

**Review Assessment: Thoroughness In Paper Reading:**

I read the paper thoroughly.

---

### Official Review · AnonReviewer2 · 2019-10-23
**Official Blind Review #2**

**Rating:** 3

**Review:**

This paper presents a universal adversarial attack, which firstly conducts existing gradient-based attacks on the sample images and then applies SVD on the perturbations from those attacks. The universal attacks are the right singular vectors of the SVD.  The experiments are conducted on attacking VGG and ResNet. In addition, theoretical analysis is also provided in the paper.

Compared with instance-wise attacks, universal attacks are relatively rare. The idea of this paper is intuitive but I feel that it is highly related to the one in Khrulkov & Oseledets (2018). The latter finds singular vectors with the gradients of the hidden layers of the targeted classifier. In general, the instance-wise attacks such as FGSM and Gradient are essentially based on gradients of the classifiers, as well. Therefore, given Khrulkov & Oseledets (2018), I would consider the novelty of this paper is not large enough, although I can see that the proposed may be more efficient.

In addition to attacking raw classifiers, I would also expect the comparisons with defence methods against universal attacks, such as the one in [1].

Minors:

It is a bit hard to compare the performance across different methods in Figure 1. I would suggest using tables to give a clearer comparison.

Overall, I think the paper stands on the borderline.

[1] Akhtar, Naveed, Jian Liu, and Ajmal Mian. "Defense against universal adversarial perturbations." Proceedings of the IEEE Conference on Computer Vision and Pattern Recognition. 2018.

**Experience Assessment:**

I have read many papers in this area.

**Review Assessment: Checking Correctness Of Derivations And Theory:**

I assessed the sensibility of the derivations and theory.

**Review Assessment: Checking Correctness Of Experiments:**

I assessed the sensibility of the experiments.

**Review Assessment: Thoroughness In Paper Reading:**

I read the paper at least twice and used my best judgement in assessing the paper.

---

### Official Review · AnonReviewer3 · 2019-10-25
**Official Blind Review #1877**

**Rating:** 3

**Review:**

This paper presents an observation that one can use the top singular vector of a matrix consisting of adversarial perturbation (Gradient attack, FGSM attack, or DeepFool) vectors of a subset of data as a universal attack (applying the same perturbation to all inputs and fools a large fraction of inputs). The paper gives a theoretical justification of their method using matrix concentration inequalities and spectral perturbation bounds.

Strengths:
- A simple and effective technique to fool a large fraction of examples leveraging the observation that only a small number of dominant principal components exist for input-dependent attack directions.

- Clean theoretical justification of the performance of the proposed methodology.

- I also like the observation and the generality, simplicity, and theoretical proof of the proposed universal attack algorithm SVD-Universal.

Weaknesses:
- Performance seems to be inferior to previous methods e.g. Khrulkov & Oseledets 2018. The paper does not give a comparison between SVD-Universal and (p,q)-SVD.

- Although the author gives a justification of why they do not compare with (p,q)-SVD, I still like to see a comparison between the two methods such that we can have a better idea about what is the potential performance loss by using the SVD-Universal when compared with (p,q)-SVD.

- It is not clear to me how the authors build the matrix corresponding to the universal invariant perturbations in sec 6.


**Experience Assessment:**

I have read many papers in this area.

**Review Assessment: Checking Correctness Of Derivations And Theory:**

I assessed the sensibility of the derivations and theory.

**Review Assessment: Checking Correctness Of Experiments:**

I assessed the sensibility of the experiments.

**Review Assessment: Thoroughness In Paper Reading:**

I made a quick assessment of this paper.

---

### Decision · Program_Chairs · 2019-12-19

**Decision:**

Reject

**Comment:**

The paper proposes to get universal adversarial examples using few test samples. The approach is very close to the Khrulkov & Oseledets, and the abstract for some reason claims that it was proposed independently, which looks like a very strange claim. Overall, all reviewers recommend rejection, and I agree with them.